# Somatic Mutations in Latin American Breast Cancer Patients: A Systematic Review and Meta-Analysis

**DOI:** 10.3390/diagnostics14030287

**Published:** 2024-01-29

**Authors:** Gabriela A. Martínez-Nava, Laura Keren Urbina-Jara, Saúl Lira-Albarrán, Henry L. Gómez, Erika Ruiz-García, María Tereza Nieto-Coronel, Rocio Ortiz-Lopez, Kenia Nadiezhda Martínez Villalba, Mariana Muñoz-Sánchez, Dione Aguilar, Liliana Gómez-Flores-Ramos, Sara Aileen Cabrera-Nieto, Alejandro Mohar, Marlid Cruz-Ramos

**Affiliations:** 1Laboratorio de Gerociencias, Instituto Nacional de Rehabilitación Luis Guillermo Ibarra Ibarra, Calz. México-Xochimilco 289, Tlalpan, Mexico City 14389, Mexico; ameria.justice@gmail.com; 2Tecnologico de Monterrey, Escuela de Medicina y Ciencias de la Salud, Monterrey 64710, Mexico; lauraqcb@gmail.com (L.K.U.-J.); rortizl@tec.mx (R.O.-L.); 3Departamento de Gestión Académica e Investigación, Hospital Escuela, Tegucigalpa 11101, Honduras; saul.lira@hospitalescuela.edu.hn; 4Departmento de Medicina Oncológica, Instituto Nacional de Enfermedades Neoplásicas, Av. Angamos Este 2520, Lima 15023, Peru; hgomez@inen.sld.pe; 5Laboratorio de Medicina Traslacional, Instituto Nacional de Cancerologia, Mexico City 14080, Mexico; betzabe100@yahoo.com.mx; 6Departamento de Medicina Oncológica, Centro Oncopalia, Universidad Mayor de San Andrés, La Paz P.O. Box 8635, Bolivia; maytemtnc2@gmail.com; 7Tecnologico de Monterrey, Institute for Obesity Research, Monterrey 64849, Mexico; 8Unidad de Epidemiología e Investigación Biomédica en Cáncer, Instituto de Investigaciones Biomédicas, UNAM-Instituto Nacional de Cancerología, Mexico City 14080, Mexico; kenia28nadiezhda@gmail.com (K.N.M.V.); mohar@iibiomedicas.unam.mx (A.M.); 9Facultad de Ciencias de la Salud, Universidad Anáhuac México, Mexico City 52786, Mexico; mariana.munozsa@anahuac.mx (M.M.-S.); sara.cabrera32@anahuac.mx (S.A.C.-N.); 10Tecnologico de Monterrey, Centro de Cáncer de Mama, Hospital Zambrano Hellion, San Pedro Garza García 66278, Mexico; oncogenetica.monterrey@gmail.com; 11CONAHCYT/Center for Population Health Research, National Institute of Public Health, Universidad No. 655, Cuernavaca 62100, Mexico; liliana.gomez@insp.mx; 12Programa Joven y Fuerte/CONAHCYT, Instituto Nacional de Cancerología, Av. San Fernando 22, Belisario Domínguez Sección 16, Tlalpan, Mexico City 14080, Mexico

**Keywords:** breast cancer, somatic mutation, exposome, target therapy, Latin America

## Abstract

(1) Background: Somatic mutations may be connected to the exposome, potentially playing a role in breast cancer’s development and clinical outcomes. There needs to be information regarding Latin American women specifically, as they are underrepresented in clinical trials and have limited access to somatic analysis in their countries. This study aims to systematically investigate somatic mutations in breast cancer patients from Latin America to gain a better understanding of tumor biology in the region. (2) Methods: We realize a systematic review of studies on breast cancer in 21 Latin American countries using various databases such as PubMed, Google Scholar, Web of Science, RedAlyc, Dianlet, and Biblioteca Virtual en Salud. Of 392 articles that fit the criteria, 10 studies have clinical data which can be used to create a database containing clinical and genetic information. We compared mutation frequencies across different breast cancer subtypes using statistical analyses and meta-analyses of proportions. Furthermore, we identified overexpressed biological processes and canonical pathways through functional enrichment analysis. (3) Results: 342 mutations were found in six Latin American countries, with the *TP53* and *PIK3CA* genes being the most studied mutations. The most common *PIK3CA* mutation was H1047R. Functional analysis provided insights into tumor biology and potential therapies. (4) Conclusion: evaluating specific somatic mutations in the Latin American population is crucial for understanding tumor biology and determining appropriate treatment options. Combining targeted therapies may improve clinical outcomes in breast cancer. Moreover, implementing healthy lifestyle strategies in Latin America could enhance therapy effectiveness and clinical outcomes.

## 1. Introduction

Breast cancer (BC) is the leading type of cancer diagnosed in Latin American women [1]. In these countries, this disease usually occurs in younger women; their tumors tend to have more aggressive clinical behavior and are commonly diagnosed in more advanced clinical stages [2]. In recent decades, Latin American women have changed reproductive behaviors and lifestyles, such as diet, physical activity, sleep patterns, stress management, and tobacco and alcohol consumption, recognized as risk factors that may increase BC incidence [3].

Based on the expression of estrogen, progesterone, HER2, and Ki67 receptors, BC can be classified into intrinsic subtypes such as luminal A or luminal B, triple-negative breast cancer (TNBC), and HER2-enriched tumors based on their gene expression profile. Latino women born or living in the United States have a higher prevalence of TNBC and HER2-enriched tumors (HR−/HER2+ and HR+/HER2−) compared to non-Hispanic white patients [4]. Thus, by frequency of presentation, in this country, the luminal A subtype is the most prevalent (67.4%), followed by TNBC (14.8%), luminal B (11.9%), and HER2+ (5.8%) for Latina women [5]. In countries such as Mexico, Colombia, Costa Rica, and Peru, TNBC has a similar prevalence of 32.1, 20.6, 17.4, and 21.3%, respectively. In contrast, a HER2+ tumor is more frequent in Peru, and the luminal A subtype is more frequent in Brazil and Colombia [5]. The luminal B and TNBC subtypes in young women with BC are most frequently found [6]. The tumor classification is relevant since somatic mutations can help us predict the disease’s clinical behavior in these populations. Most somatic mutations in BC are single-base substitutions, the most common being missense mutations. About 60% of somatic alterations are missense changes, while 5.1% result in stop codons.

The most common somatic mutations found in breast cancer worldwide, across all age groups, are *PIK3CA* (29%) and *TP53* (26%) [7]. Mutations in the gene *PI3KCA* are more frequent in HR+ tumors. Around 85% of these mutations occur in exons 9 and 20, which encode the helical and kinase domains. Most mutations are in two hotspots: the central helical domain (E542K and E545K) and the COOH terminal kinase domain (H1047R). *TP53* mutations are found in conservative regions with exons 5 to 8 and occur in only 2% of cases as hot spot mutations [8].

In Latin America, the PRECAMA study (Molecular Subtypes of Premenopausal Breast Cancer in Latin American Women) identified the highest frequency of mutations in *TP53*, *PIK3CA*, and *AKT1* (32.5%, 21.4%, and 9.5%, respectively) [9]. However, despite this effort, there is little evidence of the most frequent mutations in postmenopausal women in Latin America. These women are at increased risk for more aggressive molecular subtypes. They are diagnosed in advanced stages of the disease, possibly associated with specific somatic mutations caused by the environment and lifestyle changes. This work aimed to systematically review literature and meta-analysis on the somatic genetic burden and its tumor biology of BC in Latin America. To meet this goal, the population includes breast cancer women diagnosed in Latin American countries, studies that perform somatic mutation analysis, and data that can be used to report somatic mutation frequencies and can be included in the meta-analysis. To accomplish our objective, we incorporated research articles and case–control studies performed on breast cancer patients diagnosed in Latin American countries. The studies should include somatic mutation analysis and report the frequencies of somatic mutations that can be included in a meta-analysis.

## 2. Materials and Methods

This review conforms to the Preferred Reporting Items for Systematic Reviews and Meta-Analyses (PRISMA) statement.

### 2.1. Study Selection Process

Inclusion criteria: Databases such as PubMed, Google Scholar, Web of Science, RedAlyc, Dianlet, and Biblioteca Virtual en Salud were searched for all BC studies in 21 countries from Latin America using the following search terms “Breast cancer” and country name (“Argentina”, “Belize”, “Bolivia”, “Brazil”, “Chile”, “Colombia”, “Costa Rica”, “Cuba”, “Dominican Republic”, “Ecuador”, “El Salvador”, “Guatemala”, “Honduras”, “Mexico”, “Nicaragua”, “Panama”, “Paraguay”, “Peru”, “Puerto Rico”, “Uruguay”, and “Venezuela”). Studies published in English and Spanish between 2000 and 2020 were included in this analysis. Only 14 of 21 countries reported information on this topic, such as Argentina, Brazil, Chile, Colombia, Costa Rica, Cuba, Ecuador, Mexico, Panama, Paraguay, Peru, Puerto Rico, Uruguay, and Venezuela. Therefore, 9209 abstracts were reviewed, and 392 papers, including research articles and case report studies, reported somatic mutations for BC measured by real-time and sequencing analysis were selected to be included in our study and realize statistical analysis [9,10,11,12,13,14,15,16,17,18]. Somatic mutations analyzed by our collaborators, Dr. Gomez from Peru and Dr. Ruiz from Mexico, were also included. These authors shared their somatic mutation database with us to complement the information of our statistical analysis. The following exclusion criteria were applied to select papers for analysis: studies that only included protein or immunohistochemical analysis without any genetic information, and studies that mentioned somatic mutation in the title but included germinal mutations in their methods. Additionally, studies without clinical data for statistical analysis were excluded. Somatic BRCA mutations were also excluded from this analysis. Ten papers were included in the final analyses, as depicted in Figure 1.

### 2.2. Statistical Analysis

A database from the ten full-length articles was constructed in which clinical and genetic information of each patient was considered. The data were independently extracted and registered by reviewers (MCR, LU, DAM, LF, MM, and GAMN). The discrepancy between authors regarding whether the mutations were somatic was resolved by an additional reviewer (RO) who also extracted the data, participated in discussions with them, and made the final decision. The items included title, year, country, somatic mutation frequency, sample type, age, lymph node, histological classification and grade, tumor size, molecular subtype, estrogen, and progesterone receptor, HER2 status, Ki67, gene, chromosome, exon, intron, codon, nucleotide, protein, amino acid, mutation type, CLINVAR report, rs-number, COSMIC mutation, COSMIC pathology, COSMIC cancer.

We first performed the analyses using the mutations as the observational unit and stratified the tumor type by hormone receptor status in the following categories: HR+/HER2+, HR−/HER2+, HR+/HER2−, and TNBC. Then, absolute and relative frequencies were calculated for variables of interest, and the proportion of mutated genes between different BC tumor types was compared with Fisher’s exact test. We also compare the previously reported frequency of *PIK3CA* single-nucleotide variations (SNVs) between BC tumor subtypes with Fisher’s exact test.

Finally, we performed a meta-analysis of proportions for the genes reported in at least three papers. Pooled proportions with their respective 95% confidence intervals (CI) were estimated by random effects models using the DerSimonian and Laird method and after the Freeman–Tukey Double Arcsine Transformation, which has been demonstrated to be robust and reliable, and is the preferred transformation when performing a meta-analysis of proportions [19]. Heterogeneity between and intra-study was assessed by Cochran’s Q, I2 index, and T2 test. Potential publication bias was evaluated with DOI plots, and the LFK index was calculated. This provides a visual and numeric estimation for the asymmetry due to publication bias with a higher sensitivity than Egger’s regression [20]. All analyses and plots were performed in STATA v.14 (StataCorporation, College Station, TX, USA), and a statistical significance α level of 0.05 was considered.

### 2.3. Functional Enrichment Analysis

The functional enrichment analysis was performed using g:Profiler^β^ (version e109_e56_p17_ac9a4a5) with the g:SCS multiple testing correction method, applying a significance threshold of 0.05 [21] and uploading the list of mutated genes in every molecular subtype of breast cancer evaluated. The overrepresented biological processes were summarized with REVIGO (reduce + visualize Gene Ontology, http//revigo.irb.hr; version 1.8.1, accessed on 18 June 2023) to avoid redundant gene ontology terms [22]. The vocabulary of biological processes used the terms of the Gene Ontology Consortium [23]. In addition, the enriched canonical pathways were identified using KEGG [24], Reactome [25], and WikiPathways [26].

### 2.4. Ingenuity Pathway Analysis

Based on a hypothesis-driven approach to the loss of function mutations’ effect in genes associated with every molecular subtype of breast cancer on several bio-functions, a molecule activity predictor analysis (MAP) by QIAGEN IPA was performed (QIAGEN Inc., https://digitalinsights.qiagen.com/IPA, accessed on 19 June 2023). The Ingenuity Knowledge Base defined the bio-functions, canonical pathways, and networks evaluated [27].

## 3. Results

### 3.1. Characteristics of the Mutations Identified

A total of 342 mutations were reported in the reviewed articles. The mutations were identified in patients from six Latin American countries (Brazil, Mexico, Chile, Colombia, Peru, and Costa Rica). Most studies include early breast cancer tumors; only two studies in Brazil have metastatic patients (Table 1). Almost 80% of the mutations were found in patients from Mexico or Brazil (38.89% and 37.13%, respectively) and 7.60% from Chile (Table 2). The mutations were identified in nine genes, highlighting *TP53* and *PIK3CA* as the most screened and reported. Indeed, mutations in *TP53* represented 58.48% of the reported mutations, while mutations in *PIK3CA* accounted for 25.44%. Furthermore, other mutations in *FLT3*, *AKT1*, *CDKN2A*, *CDH1*, *PTEN*, *RB1*, and *NOTCH* represent less than 5% of the informed mutations. However, less than half of the mutations had a COSMIC identification number, and only 56.14% were linked to a reference sequence number. Unfortunately, more than half of the mutations were not reported or had no information in the ClinVar database [28], and exclusively 28.07% were classified as pathogenic. Over half were coding mutations and changed the amino acid in the encoded protein (Table 2). This type of mutation was the most prevalent for almost all reported genes, except for mutations localized in *CDH1*, where 83.33% were frameshift mutations (Appendix A). 

### 3.2. Type of Breast Cancer Tumor, Somatic Mutations Identified, and Hormone Receptor Status

From the papers where the clinical information was available, we observed that TNBC was the most frequently reported breast cancer subtype, followed by HR+/HER2−, HR+/HER2+, and HR−/HER2+ (Table 3). The distribution of reported gene mutations significantly differed between BC subtypes for most genes. *TP53* mutations were more frequently reported in the TNBC subtype (67.71%), as well as mutations in *PIK3CA* (38.57%), *FLT3* (100%), and *RB1* (80%) genes. *CDH1*, *CDKN2A*, *AKT1*, and *PTEN* mutations were more frequently found in the HR+/HER2− subtype than any other BC subtype. Nevertheless, the difference in proportions was not significant for mutations in *PTEN*. Overall, we observed that mutations in *TP53*, *PIK3CA*, *FLT3*, *AKT1*, *CDKN2A*, and *CDH1* were found in different proportions among BC subtypes (Table 3). 

Previous reports have highlighted the clinical importance of some single-nucleotide variations (SNVs) in *PIK3CA*, so we explored the frequency of these SNVs in Latin American patients. The mutation H1047R (30%) was the most reported in all subtypes, followed by E545K (11.43%), E545A (10%), and E545A (7.14%). We observed that the distribution of E545A was significantly different among BC subtypes. This mutation was most detected in HR−/HER2+ (25%) and HR+/HER2+ (50%) subtypes. In contrast, the mutation H1047R was more frequent in the HR+/HER2− (34.78%) subtype and TNBC (22.22%) (Table 4). 

We performed a meta-analysis of proportions to know the prevalence of mutations in the two most frequently reported genes in Latin American patients. Figure 2 shows the proportions estimated by each study with their 95%CI and the pooled proportion estimated by random effects models. We found that mutations in *TP53* are reported in 26% (95%CI = 0.14–0.40) of tumors, while mutations in *PIK3CA* in 19% (95%CI = 0.11–0.28) (Figure 2). The LFK index for *TP53* showed no publication bias associated with the proportion of mutations identified in these genes among Latin American patients’ tumors. However, we detected slight asymmetry for the *PIK3CA* mutations reported, meaning there was a slight publication bias among the mutations reported in this gene in Latin American patients’ tumors (Appendix A).

### 3.3. Biological Processes and Canonical Pathways Enriched in the Subtypes of Breast Cancer Tumors

The functional enrichment analysis, using the list of seven genes with mutations in the TNBC subtype (Appendix A), identified the main overrepresented non-redundant biological processes: the regulation of the apoptotic process, programmed cell death, G1/2 transition of the mitotic cell cycle, growth, and anoikis (Appendix A). In addition, phosphatidylinositol-mediated signaling, regulation of signal transduction, biogenesis, phosphorus metabolic process, and protein kinase B signaling were overrepresented (Appendix A). The top ten enriched canonical pathways defined by KEGG included breast cancer, central carbon metabolism in cancer, endocrine resistance, pathways in cancer, and cellular senescence (Appendix A). Furthermore, the sphingolipid and PI3K-Akt signaling pathways, EGFR tyrosine kinase inhibitor resistance, microRNAs in cancer, and apoptosis were identified (Appendix A). Similarly, multiple canonical pathways associated with PI3K/AKT signaling were enriched according to Reactome, together with FLT3 signaling, the regulation of TP53 activity through association with co-factors and acetylation, PTEN regulation, and oncogene-induced senescence (Appendix A). Finally, the breast cancer and integrated cancer pathways were enriched using WikiPathways, along with the PI3K-AKT-mTOR signaling pathway and therapeutic opportunities, EGFR tyrosine kinase inhibitor resistance, and the androgen and estrogen receptor signaling pathways (Appendix A).

The functional enrichment analysis using the list of eight genes with mutations in the HR+/HER2− breast cancer subtype (Appendix A) identified the main overrepresented non-redundant biological processes: the regulation of G1/2 transition of the mitotic cell cycle, growth, apoptotic process, and cell adhesion, and anoikis (Appendix A). The top ten canonical pathways enriched and defined by KEGG included endocrine resistance, pathways in cancer, breast cancer, and cellular senescence (Appendix A). Also, the central carbon metabolism in cancer, microRNAs in cancer, p53 signaling pathway, EGFR tyrosine kinase inhibitor resistance, PI3K/AKT signaling pathway, and apoptosis were identified (Appendix A). Furthermore, several canonical pathways related to TP53 were enriched according to Reactome in the HR+/HER2− breast cancer subtype (Appendix A). Finally, the breast cancer and DNA damage response (only ATM dependent) pathways were enriched WikiPathways, together with PI3K-AKT-mTOR signaling pathway and therapeutic opportunities, EGFR tyrosine kinase inhibitor resistance, apoptosis as well as the androgen and estrogen receptor signaling pathways (Appendix A). 

### 3.4. Impact of the Loss of Function Mutations in TNBC and HER-2 Breast Cancer Subtypes

The molecular activity prediction of several bio-functions defined by the Ingenuity Knowledge Base lets us hypothesize the impact of the loss of function mutations in seven genes related to the TNBC subtype (Figure 3). The proteins TP53, RB1, and PTEN are crucial in preventing tumor cells from experiencing senescence and anoikis; altering its genes contributes to cancer progression. Based on this analysis, it seems that cancer cells that have lost their attachment to the extracellular matrix and neighboring cells are hindered in undergoing apoptosis or programmed cell death. Our results highlight that the TNBC molecular subtype displays aggressive behavior due to the uncontrolled growth of cancer cells and the inhibition of apoptosis. Interestingly, the metastasis bio-function was predicted as activated mainly through the loss of function of the same three genes associated with a worse clinical prognosis in Latin American women with TNBC. In addition, five of the seven genes in this network analysis are involved in the canonical pathway: molecular mechanisms of cancer. The PI3K/AKT signaling included four genes, some of them (*AKT1*, *PIK3CA*, and *TP53*) pharmacological targets for new drugs being tested in clinical trials. In the canonical pathway, cell cycle G1/S checkpoint regulation integrated only two genes (*TP53* and *RB1*); however, the loss of function mutations’ effects was enough to predict inhibited two bio-functions associated with apoptosis.

The molecular activity prediction of several diseases defined by the Ingenuity Knowledge Base showed the effects of the loss of function mutations in eight genes related to breast cancer’s HER2− molecular subtype (Figure 4). In this sense, *TP53* and *PTEN* downregulation predict activated breast carcinoma, considering that six more genes are associated with this disease in our analysis. In contrast, *AKT1* downregulated was enough to predict inhibited breast carcinoma metastasis, correlating with the least aggressive behavior of breast cancer’s HER2− molecular subtype in the Latino population. Furthermore, six of the eight genes in this network analysis are involved in the canonical pathway: molecular mechanisms of cancer. The HER-2 signaling in breast cancer included five genes, three of them (*AKT1*, *TP53*, and *PIK3CA*) being pharmacological targets for new drugs tested in clinical trials. In the canonical pathway, cell cycle G1/S checkpoint regulation integrated three genes (*TP53*, *CDKN2*, and *RB1*); despite this, the loss of function mutations’ effects of *TP53* were sufficient to predict activated the development of breast carcinoma.

Finally, molecular activity prediction of the canonical pathway estrogen receptor signaling defined by the Ingenuity Knowledge Base included three of the six (*TP53*, *PTEN*, *NOTCH1*, *AKT1*, *PIK3CA*, and *RB1*) commonly downregulated genes with loss of function mutations in both molecular subtypes of breast cancer (TNBC and HER2−). In this regard, the downregulation of *TP53* inhibited the expression of *CDKN1A* and *PCNA*, predicting the bio-function of tumor cell proliferation as activated. *PTEN* downregulation activated the expression of Akt complex that increases the activity of NfkB, which predicted the activation of the bio-function survival of cells. On the other hand, the Akt complex inhibited the BAD expression; consequently, apoptosis is predicted to be inhibited. Interestingly, the metastasis is slightly predicted as activated based on ERK1/2 complex activity increased by MAP2K1/2-RAF1-Akt. In contrast, downregulation of *NOTCH1* decreases HES1 activity, which predicted the inhibition of the bio-functions: migration of tumor cells and invasion of breast cancer cells (Appendix A).

## 4. Discussion 

This systematic review is a look at the efforts made in Latin America to perform somatic mutation analyses in women with breast cancer that allow us to understand better the tumor biology of women living in this geographical area. Women living in Latin America have been changing their ways and customs over time, going from a lifestyle that included more physical activity and food prepared at home to now having little time to cook, living with tremendous stress, and including processed foods in their diet, which implies a change in their exposome, which comprises a greater risk of developing somatic mutations [29], as was observed in this study where somatic mutations appear since early stages. These can be observed in a recent study from The Cancer Genome Atlas (TCGA) that identified a significant association between the high body mass index (BMI) and an increase in the somatic copy number variants (SCNVs) (*p* = 0.039) in women with BC. The increased BMI was associated with driver mutations in *GATA3* (OR = 1.43, 95%CI 1.02–2.01) [30]. These results are relevant since many women in Latin America are overweight or obese and have adopted high-risk habits, such as higher alcohol consumption [29,31]. 

Although there were limitations with the available data, we found that, like other populations, mutations in both *P53* and *PI3K* were frequently reported. These findings are consistent with previous reports from other regions that include Caucasian, African-American, and Asian populations [8]. Various publications have studied the connection between p53 alteration and clinical outcomes in breast cancer. The studies indicate that the presence of *P53* mutations can lead to aggressive behavior in breast cancer and harm clinical outcomes [32]. The wild-type p53 (wtp3) protein is a genome guardian, promotes cell cycle arrest and apoptosis, and inhibits VEGF-dependent angiogenesis, tumor growth, metastasis, and drug resistance [33]. The primary mutations of *P53* occur in the DNA binding domain, with the most common being missense mutations found in exons 5 through 8. These mutations typically lead to a gain in oncogenic function, promoting invasion and metastasis, resistance to apoptosis, lack of control over the cell cycle, and genome instability. The alteration of the transcriptional regulators’ function may modify gene expression and regulatory genes, contributing to histone methylation that may lead to changes in gene expression and cancer cell growth [33]. The p53 activation intervenes multiple times in the cell, generally participating in stressful processes such as endogenous stress, replication stress, hypoxia, reactive oxygen species, oncogenic activation, exogenous stress nutrient deprivation, irradiation, and cytotoxic agents [34]. The loss of function of *P53* activates tumorigenesis and aggressive tumor phenotype and promotes invasion, angiogenesis, and drug resistance, favoring poor clinical outcomes in BC.

Mouse double minute 2 homolog (MDM2) is a protein that regulates p53 activation. When p53 accumulates into the cell, MDM2 acts as a ubiquitin ligase of the wtp53 protein, downregulating the protein levels without stress. MDM2 also prevents p53 from entering the nucleus, inhibits DNA binding, and promotes p53 proteasome degradation. PRIMA-1 (p53 reactivation and induction of mass apoptosis) is a low-molecular-weight compound that causes mutated p53 protein to undergo a conformational change and facilitates the binding to DNA in a sequence-specific manner to induce apoptosis [33]. Adding a methyl group enhances the effects of PRIMA-1 (PRIMA-1Met), also known as APR-246; this molecule is a dual MDM2 and MDMX inhibitor. As monotherapy, this drug induces cell cycle arrest and favors apoptosis with mild side effects. In BC, a combination with monoclonal antibody (2aG4), which targets exposed phosphatidylserine residues on tumor blood vessels, shows ischemia, hemorrhagic necrosis, cell death, and metastasis in HR+ [35] and TNBC [36] animal models. 

The PI3K pathway is a well-characterized signaling in breast cancer. Signaling-enhancing mechanisms include mutations of the *PI3K* gene, specifically *PIK3CA* gene mutations. PI3K protein is grouped into three classes (I–III) based on its structure and substrate specificity. Class IA PI3K has significant implications in cancer; this protein is constituted by the p110 catalytic domain and p85 regulatory domain. The catalytic domain P110 has three isoforms (α, β, and δ); p110α is encoded by *PIK3CA*, whose mutations have been reported in around 30–40% of breast cancer [37]. The majority of *PIK3CA* somatic mutations clusters in hotspots regions in exon 9 (helical domain) and exon 20 (the kinase domain), principally missense mutations change amino acid residues E542 and E545 to lysine in the helical domain (exon 9) and change H1047 to arginine in the kinase domain (exon 20), both mutations have shown to be gain-of-function mutations with transforming capacity leading to increase PI3K activity [38]. Martinez et al. studied the frequency and spectrum of *PIK3CA* somatic mutations in breast cancer according to subtype, showing that the prevalence of *PIK3CA* mutations in ER-negative/HER2-negative, HER2-positive, and ER-positive/HER2-negative breast cancer, was 18%, 22%, and 37%, respectively. In this analysis, 69% of *PIK3CA* mutations occur in exon 20: 35% in H1047R and 4% in H1047L. Exon 9 mutations in *PIK3CA* are present in 7%: E545K (17%) and E542K (11%) [39]. We found a similar mutation frequency in the Latin American population in H1047R (30%). However, H1047L was not reported in our countries. 

Mutations in the *PIK3CA* gene have been observed to be associated with resistance to chemotherapy. This suggests that tumors with *PIK3CA* mutations had fewer response rates than *PIK3CA* wild-type tumors. Notably, in TNBC, the mutation H1047R of *PIK3CA* has been linked to reduced complete tumor regression (pCR) rates when treated with anthracycline and taxane neoadjuvant chemotherapy [40]. Similarly, in HER2+ BC disease, the mutation in *PI3KCA* confers a worse response to neoadjuvant anti-HER2 therapy [41]. Particularly, H1047R mutations in *PIK3CA* in HER2+ disease are also related to metastasis risk [42] and resistance to HER2-targeted therapies (trastuzumab and lapatinib) in preclinical studies [43]. However, in other scenarios, such as metastatic hormone-positive patients, H107R mutation in *PIK3CA* may predict a response to everolimus therapy [44]. It seems that the inhibition of the PI3K pathway may overcome this resistance, so it seems that the kind of mutation and subtype are important to predict clinical outcomes on BC [43,45], making these mutations useful for identifying more aggressive breast cancer tumors and predicting the response to PI3K pathway inhibitors [11].

In HR+/HER2− metastatic tumors, the *PI3KCA* mutations are present in around 40% of cases, and endocrine therapy and cyclin-dependent kinase 4 and 6 (CDK4/6) inhibitors are the standards of care to treat these tumors. *PI3KCA* mutations are involved in CDK4/6 response [46]. In the MONALESA trial, the median number of months of progression-free survival is 19.2 months for the mutated *PIK3CA* group and 29.6 months for the wild-type *PIK3CA* group. However, *PI3KCA* mutations do not significantly affect clinical outcomes [47]. *PI3KCA*-mutant patients treated with abemaciclib in the MONARCH 3 study achieved a median PFS of 27 months, while in the wild-type group, the median survival has not reached [48]. In the PALOMA-3 study, the *PI3KCA* mutation detected by circulating DNA at the study enrollment shows no significant difference in recurrence-free survival between the mutated and non-mutated groups [49]. However, after the chronic exposition to CD4/6 inhibitors, the role of the *PI3K* mutations has a different behavior. PI3K/mTOR has been reported to be upregulated in response to this exposition, and as an effect, cyclin D is upregulated and activates CDK2, which subsequently drives cell cycle progression. This has been reported by the PALOMA-3 study [49], where the analysis of circulating DNA sequencing demonstrated driver mutations in *PIK3CA* and *ESR1*. Those patients with greater exposure to CD4/6 inhibitors appeared more likely to develop driver gene mutations [50]. However, further studies are also required to be more specific in the exon and specific mutation to determine the fundamental role of *PI3KCA* mutation on resistance to CDK4/6 inhibitors. For example, *PIK3CA* mutation on H1047R is associated with higher clinical benefit from alpelisib than mutations in the helical domain [51].

PI3K inhibitors emerge as a good alternative to overcome resistance to hormonal and cyclin inhibitors resistance. When we analyzed the studies with buparlisib, pictilisib, alpelisib, and taselisib, the mutated *PI3KCA* group had a better response than the wild-type group in such a way that the evaluation of the *PI3KCA* mutation is a response biomarker for this group of drugs [52]. Furthermore, phase III SOLAR-1 (Clinical Studies of Alpelisib in Breast Cancer 1) evaluates the safety and efficacy of alpelisib and α-specific class-1 PI3K inhibitor plus fulvestrant in HR+HER2− metastatic breast cancer patients exposed to endocrine therapy. Alpelisib positively resulted in progression-free survival (11 months vs. 7.4 months), overall response, and clinical benefit vs. placebo/fulvestrant [53]. 

Strategies to overcome resistance to treatment is the combination of PI3KCA inhibitors and other therapies, which may improve clinical results in BC based on molecular profile. Ongoing trials are evaluating combination of ribociclib, fulvestrant + buparlisib (NCT02088684), palbociclib + taselisib + fulvestrant or letrozole (NCT02389842), abemaciclib + fulvestrant + copanlisib (NCT03939897), palbociclib+ + gedatolisib or palbociclib + fulvestrant + gedatolisib (NCT02684032), copanlisib + palbociclib (NCT03128619), and ribociclib + buparlisib + letrozole (NCT02154776) [52].

In TNBC, a pharmacological combination may be a therapeutic future option. Lehman et al. [54] described that AR+ TNBC cell lines are resistant to cisplatin, which reinforces the belief that patients with AR+ TNBC tumors may not have the same clinical benefit from standard chemotherapy as patients with TNBC without androgen receptors and would benefit from therapeutic combinations that simultaneously target AR and PI3K. In TNBC subtypes, the most frequent clonal events are *TP53* mutations (62%), followed by mutations in *PIK3CA* (10.2%). In this regard, 40% of *PIK3CA* mutations in the AR+ TNBC receptor subtype have been reported vs. TNBC without androgen receptors (4%) [54], so PI3K inhibitors may be an alternative to these patients. In patient-derived xenografts (PDX) of AR+ TNBC, *PIK3CA* mutations were strongly correlated with AR+ TNBC [38% (8/21) vs. 10% (30/302) in other subtypes; *p* = 0.001)] [54]. These findings suggest combining AR antagonism and PI3K inhibition for the additive or synergistic effect on AR + TNBC cell growth. In addition, preclinical results provide a strong rationale for using AR as a biomarker for selecting TNBC patients for clinical trials, which would investigate the efficacy of therapeutic combinations that simultaneously target AR and *PI3K.* For instance, one clinical trial Phase 1 (NCT03207529) evaluates alpelisib + enzalutamide in AR-positive and *PTEN*-positive metastatic BC [55].

Regarding exon 9 mutations, our study found that E545K and E542K are mutated in less proportion than other populations. However, E545A mutation is present in 10% of the patients included in this study, which is higher than in other reports where the frequency is around 0.5% [39]. In Peru, 15.7% of *PIK3CA* mutations in BC have been reported in TNBC and the HER2+ tumors, where the E545A mutation is the most frequent [15]. A recent study in Brazil also shows more frequency of E545A mutation, especially in young patients with BC [56]. The exon 9 mutations in *PIK3CA* in breast cancer enable p110α to escape from the inhibitory effect of p85, promoting constitutive PI3K signaling. The mutations in exon 9 are associated with more aggressive disease in the Caucasian population [57], which may explain the aggressive behavior in Latin women with breast cancer. However, further research is required to understand the role of exon 9 mutations comprehensively. 

*PTEN* loss of expression is another condition of resistance to chemotherapy [58], anti-HER2 therapy prescription [59], and alpelisib response in BC [60]. PTEN regulates the PI3K/AKT pathway, and the loss of function enhances its signaling. Based on the molecular activity prediction analysis, the *PTEN* loss decreases the anoikis and the senescence but increases the metastasis. In our systematic revision, we found the loss of *PTEN* in a lower proportion, principally in HR+ and TNBC; this loss is related to the hyperactivation of PI3K/AKT signaling. In this regard, pictilisib [61], a pan inhibitor that targets PI3Kβ, showed a benefit independent of *PTEN* status in the OPPORTUNE trial [62]. Other PI3K/AKT members, such as *AKT*, were identified as mutated in our analysis; however, the frequency occurs in less proportion than other populations, around 7% [63]. As in other studies, *AKT* mutation is present in a significant proportion of HR+ tumors and may be involved in a high risk of metastasis in this group. Ongoing clinical trials with AKT inhibitors are testing the efficacy of these drugs in BC [64]. However, more studies are needed to evaluate the hyperactivation of PI3K/AKT in the Latino population to understand the aggressive behavior or BC in Latin America and predict response to the standard therapy.

The Latino population has a high incidence of diabetes, and unfortunately, some drugs, such as the PI3K inhibitors used in the pharmacological treatment of breast cancer, lead to hyperglycemia. This condition increases insulin release and reactivates PI3K signaling in animal models [65]. On the other hand, the SANDPIPER trial shows evidence of differences in taselisib efficacy according to the world region; the hazard ratio (HR) was 0.38 (95%CI 0.19–0.75) in Asia, 0.57 (95%CI 0.41–0.79) in Western/Europe/USA/Canada/Australia and 1.18 (95% CI 0.78–1.77) in Latin America /Eastern Europe [66]. These differences can be related to metabolic syndrome, diet, and glycemia control in these regions [65]. Latin American populations may require strategies for a healthy lifestyle to improve the efficacy of BC therapies.

Our analysis has significant limitations due to insufficient clinical data founded in the studies included in this analysis. Although we have found some sequencing analyses and targeted gene studies, we need more data on the Latin American population. As per the TCGA Data Portal, less than forty cases meet the criteria for breast cancer diagnosis in female Hispanic or Latino women [67]. Also, there needs to be more information on mutations in estrogen receptor that are also important to predict treatment resistance and clinical outcomes.

Although there is some information about *TP53* and *PI3K* mutations from clinical studies, it is essential to know more about the value of the specific mutations found and their effect on the treatment response. For example, in the case of mutations in *PI3K*, it is necessary to know the weight of each of its mutations, carrying out clinical studies with specific analyses that allow us to evaluate which inhibitor or drug combination is best in each case. This would save the use of drugs that are not useful in these patients if there is no response to treatment. 

Likewise, CD4/6 inhibitors have recently proven to be helpful in adjuvant treatment in the case of patients with high-risk HR+ BC, in studies like monarchE [68] and NATALLIE [69]. Therefore, questions arise as to whether this strategy will develop mutations that could restrict or favor the use of these or CD4/5 and PI3K inhibitors in the event of relapse or progression. The design of the studies based on the molecular analysis of the patient seems to be of utmost importance in the pursuit of answers [70].

On the other hand, although sequencing techniques are currently more accessible in Latin America, not all patients will have access to this technique. If they do, this does not guarantee that they will have access to the medication. More studies in Latin America, such as Brazil [71] and Mexico, need to be carried out to show that drug access improves patients’ clinical outcomes. Indeed, if molecular analysis is added, the resource can be optimized more appropriately.

It is important to note that studies have shown that dietary and exercise interventions can potentially modify the active insulin levels in the PI3K/AKT pathway. Specifically, the mTOR protein and its upstream activators (PI3K, AKT, Ras) in skeletal muscle cells are responsive to acute exercise and feeding (amino acids) and regulate protein synthesis and cell growth. Interestingly, research has also found that acute exercise-conditioned serum can reduce phosphorylation of AKT, ERK 1/2, mTOR, and p7S6K [72]. Further molecular analysis is needed to fully understand the effects of lifestyle interventions in breast cancer and their impact on overcoming resistance mechanisms and escaping oncology treatment. These interventions may prove to be more sustainable and reduce the risk of chronic diseases in breast cancer patients.

Based on studies presented in this study that include drug response and toxicity, somatic mutation analysis seems crucial for BC patients. However, this analysis should be conducted early to select appropriate drugs and prevent side effects. More studies are needed that include somatic mutations from the early stages to understand the biology of BC and the differences in response and toxicity at these stages with a somatic mutation. Additionally, future studies that analyze somatic mutations and lifestyle interventions are also required to evaluate the change in mutation expression with these kinds of feasible and accessible interventions in the majority of populations around the world.

## 5. Conclusions

It is necessary to assess somatic mutations in Latin American women with breast cancer to gain insight into the tumor biology specific to this population. According to the study results, somatic mutations occur in this population from an early stage. Target therapies based on molecular findings for breast cancer are being tested in clinical trials. A combination of drugs seems to be the best way to improve breast cancer therapy. Recently, clinical trials in the adjuvant setting have shown that combining targeted agents with standard treatment improves clinical outcomes. Specific mutations, such as E545A in *PI3KCA*, are more prevalent in other populations. The importance of these mutations in a clinical setting has not been fully explored. Latin American countries must prioritize research in this area to determine the optimal treatment options and sequence for this population. This will justify the investment made by these countries in terms of finances and resources. Lifestyle interventions are needed to control metabolic syndrome and improve standard therapy efficacy, such as PI3K inhibitors. Still, its interventions are accessible for low-income countries and have a role in epigenetics and the internal exposome of breast cancer women. 

## Figures and Tables

**Figure 1 diagnostics-14-00287-f001:**
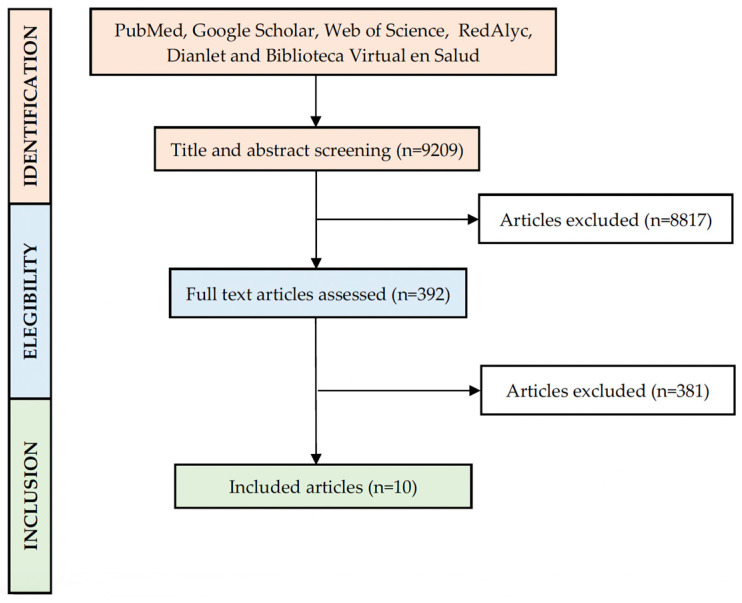
PRISMA flow diagram of the study selection.

**Figure 2 diagnostics-14-00287-f002:**
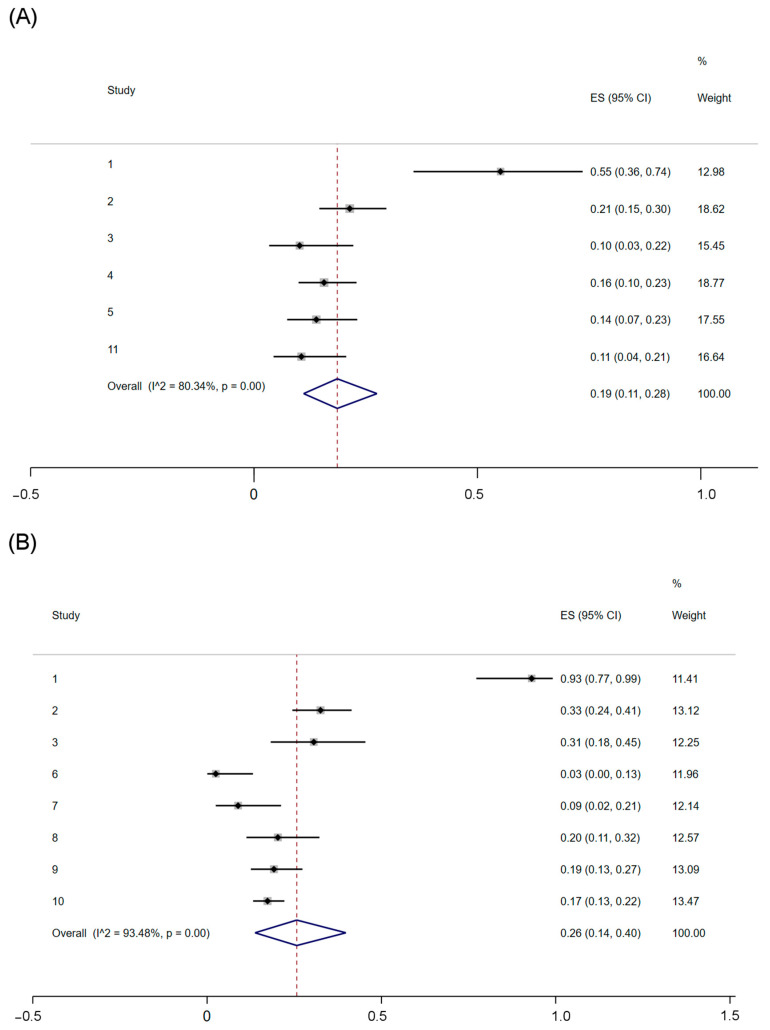
Meta-analyses of proportions by random effects models of mutations identified in (**A**) *PIK3CA* and (**B**) *TP53* estimated by random effects models using the DerSimonian and Laird method and after Freeman–Tukey Double Arcsine Transformation.

**Figure 3 diagnostics-14-00287-f003:**
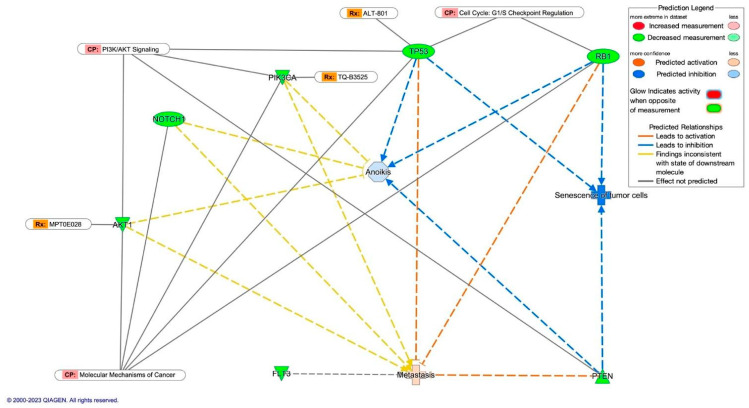
Molecular activity prediction analysis of the bio-functions metastasis, anoikis, and senescence of tumor cells related to the TNBC subtype. The figures show the downregulation of *TP53*, *RB1*, and *PTEN* secondary to loss of function mutations. This mutation predicts activated metastasis and inhibits tumor cells’ anoikis and senescence. CP = canonical pathway. Rx = drug being tested in clinical trials.

**Figure 4 diagnostics-14-00287-f004:**
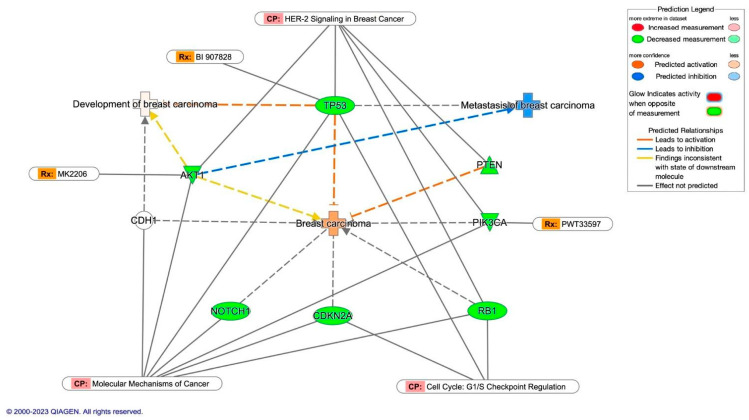
Molecular activity prediction analysis of breast carcinoma, development of breast carcinoma, and metastasis of breast carcinoma in HER2− molecular subtype. The figure shows the downregulation of *TP53* and *PTEN* secondary to loss of function mutations predicts the development of breast carcinoma. The downregulation of *AKT1* was enough to predict inhibited breast carcinoma metastasis. CP = canonical pathway. Rx = drug being tested in clinical trials.

**Table 1 diagnostics-14-00287-t001:** Clinical characteristics of breast cancer patients from Latin America were included in the meta-analysis.

Country	Molecular Subtype/Histology	*N*	Age Years (Median, Range)	Clinical Stage	Reference
Precama Study (Chile, Brazil, Mexico)	HR+ HER2+ TNBC	126	20–45	EBC	[9]
Brazil	HR+ HER2+ TNBC	32	61	EBC	[10]
Brazil	HR+ HER2+ TNBC	86	55	EBC (*n* = 37) Metastatic (*n* = 49)	[11]
Brazil	N/A	45	N/A	N/A	[12]
Brazil	HR+ TNBC	294	<50, 100 >50, 142	EBC (*n* = 269) Metastatic (*n* = 21)	[13]
Brazil	HR+ TNBC	64	53	EBC	[14]
Peru	HR+ HER2+ TNBC	134	<50, 70 ≥50, 60	EBC	[15]
Mexico	TNBC	29	51	EBC	[16]
Brazil	HR+ TNBC	58	59	EBC	[17]
Brazil	Medullar	40	N/A	EBC	[18]

**Table 2 diagnostics-14-00287-t002:** General description of the somatic mutations found in patients from Latin America.

	Frequency	
	*N* = 342	%
**Country**		
Brazil	133	38.89
Mexico	127	37.13
Chile	26	7.60
Colombia	24	7.02
Peru	21	6.14
Costa Rica	11	3.22
**Gene**		
*TP53*	200	58.48
*PIK3CA*	87	25.44
*FLT3*	16	4.68
*AKT1*	12	3.51
*CDKN2A*	7	2.05
*CDH1*	6	1.75
*PTEN*	6	1.75
*RB1*	5	1.46
*NOTCH1*	3	0.88
**Mutation type**		
Missense	191	55.85
Intronic	58	16.96
Frameshift	31	9.06
Nonsense	25	7.31
Synonymous	20	5.85
Deletion	9	2.63
Insertion	4	1.17
In frame	2	0.58
Stop codon lost	1	0.29
Uncategorized	1	0.29
**COSMIC ID**		
Yes	166	48.54
Unknown	2	0.58
Not reported	99	28.95
Without information	75	21.93
**Reference sequence ID**		
Yes	192	56.14
No	74	21.64
Without information	76	22.22
**Clinvar**		
Not reported	89	26.02
Pathogenic	96	28.07
Drug response	22	6.43
Benign	23	6.72
Uncertain	16	4.68
Without information	91	26.61
Conflict interpretation	8	2.34

**Table 3 diagnostics-14-00287-t003:** Reported somatic mutation by gene and breast cancer subtype in patients from Latin America.

Gene	Frequency (%) in a Molecular Subtype of Breast Cancer				*p*-Value *
	HR+/HER2+ (*n* = 20)	HR−/HER2+ (*n* = 11)	HR+/HER2− (*n* = 69)	TNBC (*n* = 115)	
** *TP53* **	7 (7.29)	3 (3.13)	21 (21.88)	65 (67.71)	0.002
** *PIK3CA* **	12 (17.14)	8 (11.43)	23 (32.86)	27 (38.57)	2.8 × 10^−4^
** *FLT3* **	0 (0.00)	0 (0.00)	0 (0.00)	15 (100.00)	0.002
** *AKT1* **	0 (0.00)	0 (0.00)	10 (90.91)	1 (9.09)	0.001
** *CDKN2A* **	0 (0.00)	0 (0.00)	4 (100.00)	0 (0.00)	0.048
** *CDH1* **	0 (0.00)	0 (0.00)	5 (100.00)	0 (0.00)	0.023
** *PTEN* **	0 (0.00)	0 (0.00)	4 (66.67)	2 (33.33)	0.420
** *RB1* **	0 (0.00)	0 (0.00)	1 (20.00)	4 (80.00)	0.841
** *NOTCH1* **	1 (33.33)	0 (0.00)	1 (33.33)	1 (33.33)	0.407

* *p*-value obtained from Fisher’s exact test.

**Table 4 diagnostics-14-00287-t004:** Reported somatic mutations in *PIK3CA* by breast cancer subtype in patients from Latin America.

*PIK3CA* Mutation	Frequency (%)					
	TOTAL (*n* = 70)	HR+/HER2+ (*n* = 12)	HR−/HER2+ (*n* = 11)	HR+/HER2− (*n* = 69)	TNBC (*n* = 115)	*p*-Value *
**H1047R**	21 (30.00)	4 (33.33)	3 (37.50)	8 (34.78)	6 (22.22)	0.716
**E545K**	8 (11.43)	1 (8.33)	1 (12.50)	5 (21.74)	1 (3.70)	0.205
**E545A**	7 (10.00)	3 (25.00)	4 (50.00)	0 (0.00)	0 (0.00)	3.6 × 10^−5^
**E542K**	5 (7.14)	2 (16.67)	0 (0.00)	2 (8.70)	1 (3.70)	0.420

* *p*-value obtained from Fisher’s exact test.

## Data Availability

The dataset analyzed in the current study is available from the corresponding author on reasonable request.

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
