# Peer review of "Somatic Mutations in Latin American Breast Cancer Patients: A Systematic Review and Meta-Analysis"

_diagnostics, 2024, doi:10.3390/diagnostics14030287_

Round 1
Reviewer 1 Report
Comments and Suggestions for Authors
Authors need to address the following suggestions.
1. Inclusion and exclusion criteria need to be clearly presented.
2. Limitations of this review need to be consolidated and presented at the end .
3. Discussion section must highlight the current research gaps related to Somatic Mutations in Breast Cancer Patients
4. Future directions need to be presented clearly.
5. Figures lack clarity and clear interpretation in the description .
6. Materials and methods section is highly limited. It does not explain the merits and demerits of the discussed methods.
7. A systematic review must highlight the method details, details of the samples, related statistics in a table. Authors need to address this concern and present the tabulated findings.
Comments on the Quality of English LanguageMinor changes required
Author Response
Dear reviewer,
We express our sincere gratitude for your constructive comments. Your opinion holds great value for us, and we have taken the time to address each of your remarks.
-
Inclusion and exclusion criteria must be clearly presented.
Thank you for your valuable contribution. We highly appreciate your feedback, and in response, we have meticulously refined the description of the inclusion and exclusion criteria to enhance accuracy and clarity for our readers.
-
The limitations of this review should be consolidated and presented at the end.
We are grateful for your comments, and we thank you for bringing this matter to our attention. In response, we have relocated the limitations to a dedicated section at the conclusion of the manuscript.
-
The discussion section should highlight current gaps in research related to somatic mutations in breast cancer patients.
Thanks for pointing out this aspect. Your insights are invaluable, and in recognition of your input, we have incorporated additional research gaps related to somatic mutations in breast cancer patients into the discussion.
-
Future directions need to be clearly presented.
We extend our gratitude for your significant contribution. We are actively engaged in clarifying the future directions to ensure a more lucid presentation in this section.
-
The figures lack clarity and a clear interpretation in the description.
We sincerely appreciate your keen observation. Your comments are valued, and in response, we have expanded the legend of Figure 2 to provide a more comprehensive description of the methodology used and its interpretation in the results section.
-
The materials and methods section is very limited. It does not explain the advantages and disadvantages of the methods discussed.
Thank you for bringing this to our attention. Your insightful comments are highly appreciated, and we have responded by expanding the statistical methods section to elucidate the primary reasons and merits of the methods employed in the conducted meta-analysis.
-
A systematic review should highlight method details, sample details, and related statistics in a table. The authors should address this concern and present tabulated results.
We appreciate your insightful suggestion. Your feedback is highly valued, and to address this concern, we have included the clinical data and specific details of mutations in Tables 1 and 2, previously found in Supplementary Tables 1 and 2. This adjustment aims to present a clearer description of the details and estimated proportions of mutations reported in the included studies.
Once again, we thank you for your thoughtful feedback, which has significantly contributed to the refinement of our work.
Reviewer 2 Report
Comments and Suggestions for Authors
Esteemed authors and editorial team,
I find the research theme of this article very important since designing a genetic map of breast cancer mutations should be one of our preoccupations. This would allow us the better undertanding of specific patterns and tailored management of cases.
The article is quite well written, I have only a couple of suggestions.
I would keep the introduction simpler, amd move the statistical reports to the discussions sector.
The references are quite outdated, with the exception of those tackling potential therapies. I am sure some of them could be changed.
Those who could not, such as data reporting Latin America genetic statistics, should be commented as such. They are all studies prior to 2015. Even Latina women lifestyle changes impacting the exposime are surely more recent, therefore analysis performed more than 10 years before probably do not reflect these changes.
Author Response
We express our gratitude for your thoughtful comments. In response to your insights, we have made refinements to our manuscript. The introduction has been succinctly restructured, with a heightened focus on somatic mutations in breast cancer. Furthermore, specific information has been relocated to the discussion section for improved coherence.
While we acknowledge that certain references may appear dated, it is important to note that our search methodology was grounded in the mutations prevalent within our studied population and the corresponding potential therapeutic targets for these specific mutations. This contextualization may explain the inclusion of references that may not be the most recent. To elucidate potential therapeutic targets addressing these particular mutations, a comprehensive literature review has been revisited, leading to the inclusion of pertinent and current references.
We appreciate your diligence in reviewing our work and trust that these adjustments contribute to the overall clarity and relevance of our manuscript.
Reviewer 3 Report
Comments and Suggestions for Authors
The early identification of mutational landscape of breast cancer patients is vital to select the most effective therapy for the specific patient. The authors have reviewed the papers devoted to germ mutations in breast cancer patients in Latin America : Urbina-Jara, L.K.; Rojas-Martinez, A.; Martinez-Ledesma, E.; Aguilar, D.; Villarreal-Garza, C.; Ortiz-Lopez, R. Landscape of Germline Mutations in DNA Repair Genes for Breast Cancer in Latin America: Opportunities for PARP-Like Inhibitors and Immunotherapy. Genes 2019, 10, 786. https://doi.org/10.3390/genes10100786. The current and previously published paper were conducted using publications that provide genetic and clinical data that meet the criteria for creating a database of clinical genetic studies.
Though the frequency was estimated separately for germline and somatic mutations, breast cancer (BC) patients may carry germline and somatic mutations as was shown for in the BRCA1 and BRCA2 genes, that has important clinical consequences ( C. Winter, M.P. Nilsson, E. Olsson, A.M. George, Y. Chen, A. Kvist, T. Törngren, J. Vallon-Christersson, C. Hegardt, J. Häkkinen, G. Jönsson, D. Grabau, M. Malmberg, U. Kristoffersson, M. Rehn, S.K. Gruvberger-Saal, C. Larsson, Å. Borg, N. Loman, L.H. Saal Targeted sequencing of BRCA1 and BRCA2 across a large unselected breast cancer cohort suggests that one-third of mutations are somatic. Annals of Oncology, V 27, Issue 8, 2016, Pages 1532-1538)
“Identification of an incidental BRCA mutation in cancer patients still has clinical utility”. (Shawn Yost, Elise Ruark, Ludmil B Alexandrov, Nazneen Rahman, Insights into BRCA Cancer Predisposition from Integrated Germline and Somatic Analyses in 7632 Cancers, JNCI Cancer Spectrum, Volume 3, Issue 2, June 2019, pkz028, https://doi.org/10.1093/jncics/pkz028. “ BRCA1 and BRCA2 are included in the genes recommended by ACMG to be returned if an incidental mutation is identified” (S.S.Kalia, K. Adelman, S.J. Bale, W.K. Chung, C.Eng, J.P.Evans, et al. Recommendations for reporting of secondary findings in clinical exome and genome sequencing, 2016 update (ACMG SF v2.0): A policy statement of the American College of Medical Genetics and Genomics Genet Med, 19 (2) (2017), pp. 249-255).
Two-thirds of the BRCA1 mutations found in BC are germline, and the remaining third relates to somatic mutations : Loboda AP, Adonin LS, Zvereva SD, Guschin DY, Korneenko TV, Telegina AV, Kondratieva OK, Frolova SE, Pestov NB, Barlev NA. BRCA Mutations-The Achilles Heel of Breast, Ovarian and Other Epithelial Cancers. Int J Mol Sci. 2023 Mar 5;24(5):4982. doi: 10.3390/ijms24054982. PMID: 36902416; PMCID: PMC10003548.
Meric-Bernstam F., Brusco L., Daniels M., Wathoo C., Bailey A.M., Strong L., Shaw K., Lu K., Qi Y., Zhao H., et al. Incidental Germline Variants in 1000 Advanced Cancers on a Prospective Somatic Genomic Profiling Protocol. Ann. Oncol. 2016;27:795–800. doi: 10.1093/annonc/mdw018.
Tutt A., Tovey H., Cheang M.C.U., Kernaghan S., Kilburn L., Gazinska P., Owen J., Abraham J., Barrett S., Barrett-Lee P., et al. Carboplatin in BRCA1/2-Mutated and Triple-Negative Breast Cancer BRCAness Subgroups: The TNT Trial. Nat. Med. 2018;24:628–637. doi: 10.1038/s41591-018-0009-7.
Winter C., Nilsson M.P., Olsson E., George A.M., Chen Y., Kvist A., Törngren T., Vallon-Christersson J., Hegardt C., Häkkinen J., et al. Targeted Sequencing of BRCA1 and BRCA2 across a Large Unselected Breast Cancer Cohort Suggests That One-Third of Mutations Are Somatic. Ann. Oncol. 2016;27:1532–1538. doi: 10.1093/annonc/mdw209.
The question is –though the most frequent somatic mutations reported were p53, PIK3CA ,ACT1 etc, were the BRCA1/2 SOMATIC mutations found, were they looked for?
Are there data available for the patients on both germline and somatic mutations, allowing to estimate the correlation between the most important germline mutations and the frequency of somatic mutations?
Author Response
Thank you for your feedback. The exclusion of BRCA from this study was a deliberate decision due to its integration into an independent work currently undergoing review by another journal. Notably, Dr. Laura is one of our lead authors for the first article.
As highlighted by Urbina-Jara and collaborators, the available information on Breast Cancer (BC) in Latin America is robust concerning germline data. Notably, only 11 out of 21 Latin American countries (52%) possess germline data for BC. Among these, Argentina, Brazil, Mexico, and Chile have contributed significantly, publishing the majority of articles, while other nations have limited or no available information.
For BC in Latin America, a total of 818 germline variants were identified, with 692 (84.6%) variants attributed to BRCA1/2 genes and 126 (15.4%) variants in 43 non-BRCA genes. Specifically, BRCA2 exhibited 363 variants, and BRCA1 accounted for 329 variants. Significantly, pathogenicity remains unclear for 411 (50.2%) variants, encompassing Variants of Uncertain Significance (VUS) and variants not reported in databases.
Non-BRCA genes, including ATM, TP53, CHEK2, BARD1, MLH1, PALB2, and BRIP1, displayed the highest frequency of reported germline variants in Latin America. There is a discernible imperative to define the prevalence and pathogenicity of genes associated with BC in this region. Somatic data for BC are notably scarce, necessitating further studies to facilitate a comparative analysis of the germline and somatic mutational landscape in our locale.
Your attention to these aspects is greatly appreciated, and we trust that these refinements enhance the clarity and precision of our manuscript.
Round 2
Reviewer 1 Report
Comments and Suggestions for Authors
Authors have addressed my suggestions
Comments on the Quality of English LanguageNone